# The Threat of COVID-19 on Food Security: A Modelling Perspective of Scenarios in the Informal Settlements in Windhoek

Ndeyapo M. Nickanor [1], Godfrey Tawodzera [2] and Lawrence N. Kazembe [1,*]

[1] Department of Statistics and Population Studies, Faculty of Science, University of Namibia, Windhoek 13301, Namibia
[2] Institute for Social Development (ISD), University of the Western Cape, Bellville 7535, South Africa
*   Correspondence: lkazembe@yahoo.com

**Abstract:** Due to the heterogeneity among households across locations, predicting the impacts of stay-at-home mitigation and lockdown strategies for COVID-19 control is crucial. In this study, we quantitatively assessed the effects of the Namibia government's lockdown control measures on food insecurity in urban informal settlements with a focus on Windhoek, Namibia. We developed three types of conditional regression models to predict food insecurity prevalence (FIP) scenarios incorporating household frequency of food purchase (FFP) as the impacting factor, based on the Hungry Cities Food Matrix. Empirical data were derived from the 2017 African Food Security Urban Network (AFSUN) Windhoek study and applied univariate probit and bivariate partial observability models to postulate the relation between food insecurity and FFP within the context of stay-at-home disease mitigation strategy. The findings showed that FFP was positively correlated with the prevalence of food insecurity (r = 0.057, 95% CI: 0.0394, 0.085). Daily purchases portrayed a survivalist behaviour and were associated with increased food insecurity (coeff = 0.076, *p* = 0.05). Only those who were purchasing food twice monthly were significantly associated with reduced food insecurity (coeff = −0.201, *p* = 0.001). Those households in informal settlements were severely impacted by food insecurity (coeff = 0.909, *p*-value = 0.007). We conclude that public health compliance should precede with cash or food support to poor households in balance with the need for long-term placement of control measures to fully contain COVID-19 or related infectious diseases.

**Keywords:** food insecurity; food purchase frequency; COVID-19; partial observability model; Hungry Cities Food Matrix; Namibia

## 1. Introduction

A novel coronavirus responsible for COVID-19 respiratory disease was first reported in Wuhan, China in December 2019 [1,2]. Despite numerous containment efforts, the virus continues to spread across the globe, making it a truly global pandemic. As of 23 February 2023, approximately 674-million confirmed COVID-19 cases and close to 6.87-million deaths were recorded [3]. The disease's coverage and epidemiology are still unfolding, with between 163,000 and 293,000 confirmed daily cases added to the global total cases at the peak of the epidemic between 2020 and 2021 [3]. Currently, COVID-19 has no cure, and its severity is generally above that of seasonal influenza or the H1N1 (2009) influenza [4], thus increasing the need for a vaccine. While the world was waiting for the vaccine to be developed [3,5], countries had to contend with putting in place measures to minimize the spread of the disease.

Countries responded differently to the pandemic in order to contain or mitigate the spread of the virus. Most deployed various combinations of measures that include the tracing and isolation of contacts, social-distancing, hand sanitization, and wearing of face masks. In many cases, these measures are accompanied by stringent lockdowns [6–8]. The

heterogeneity of the implemented strategies across countries reflected the fact that the optimal approach was unknown and was context-dependent.

Like elsewhere, in response to the confirmation of the first three cases of COVID-19 in Namibia, on 14 March, non-pharmaceutical interventions (NPIs) were implemented, initially implementing less-stringent measures in line with health surveillance. This mainly included isolation, contact tracing and testing, school closures, and the cancellation of sports activities. From 24 March 2020, all entries into the country were banned and any returning nationals were subjected to a 14-day self-quarantine. The discovery of further 11 COVID-19 cases resulted in stringent measures being applied, culminating into a lockdown of the Khomas and Erongo regions. The lockdown, which initially came into effect on 27 March, was extended by two weeks to cover the remaining 12 regions in the country. Announced on 14 April, this expanded phase of the lockdown ended on 4 May 2020.

As observed elsewhere in the global economy, lockdown measures had detrimental impacts on economic activities [9–13]. Some of the impacts included high levels of unemployment and shrinking economies [14,15]. This suggests that infectious diseases are a major threat to both the health and economic well-being of people around the world. The literature on the transmission of infectious diseases, particularly on viruses that have significantly impacted developing countries, highlights that the highest impact areas are generally those that have low income, poor sanitary conditions, and poor health care conditions. This has especially been the case for the spread of the Zika virus [16] and for the recent Ebola virus [17]. Other authors [18] used Ebola to develop an Infectious Disease Vulnerability Index for countries in Africa. There is now abundant literature on the socioeconomic determinants of the spread of infectious diseases in developed countries [19]. Moreover, there is more data available that suggest strong links between the spread of diseases and how their containment compromised well-being and food security [20,21]. More specifically, preliminary insights into the socio-economic impact of COVID-19 indicated that the disease will undoubtedly impact food security by interfering with food production and the smooth operation and resilience of food systems in general, as well as impinging on people's livelihood activities and, therefore, their ability to access and procure food [22–24]. Be as it may, there is no study that we are aware of that clearly demonstrated the potential effects of COVID-19 containment measures on food security. As COVID-19 continues to ravage the globe, although at an abated pace, it is certain that the food security of most people, particularly the poor, will be negatively impacted for some time to come. The need to predict different food security scenarios for planning purposes is, therefore, self-evident.

*COVID-19 Containment Stringency and Food Insecurity*

Although the COVID-19 pandemic surfaced in Africa much later than in other parts of the world, it is across Africa where experts believe the effects were felt the hardest [25,26]. This is primarily because labour shortages and price fluctuations, combined with stringent government measures that are restricting movement and trade, are likely to have significant impacts on food security across the continent. According to the World Bank, the COVID-19 pandemic is likely to cause a decline of between 2.5 and 5% in economic growth for sub-Saharan Africa in 25 years, a reduction in agricultural production, and a decrease in food imports as a result of a global recession triggered by the COVID-19 pandemic [27]. As the coronavirus crisis unfolds, disruptions in domestic food supply chains and other shocks affecting food production, as well as a loss of incomes and remittances, are creating strong tensions and food security risks in many countries.

Labour shortages due to morbidity, movement restrictions, and social distancing rules are starting to impact producers, processors, traders, and trucking/logistics companies in food supply chains—particularly for food products that require workers to be in close proximity. At the same time, a loss of income and remittances is reducing people's ability to buy food and compensate farmers for their production. The United Nations World Food Programme, jointly with FAO, IFAD, and the World Bank has warned that an estimated 265-million people could face acute food insecurity by the end of 2020, up from the

135-million people predicted before the crisis [27,28]. Food security "hot spots" include, among others, the poor and vulnerable, including the more than 820-million people who were already chronically food insecure before the COVID-19 crisis impacted movement and incomes.

This paper assessed the socioeconomic conditions that were manifesting in the area of high poverty, particularly focusing on food insecurity in informal settlements of Windhoek, Namibia. Like others in the sub-Saharan Africa region, Namibia is a country whose demographic composition differs from that of highly impacted developed countries. Africa is urbanizing rapidly and is not able to cope with this growth. By 2050, Africa's urban population is expected to increase by 1.4-billion people. The level of urbanization in Southern Africa is estimated at 64%, with countries such as Botswana (69%), Namibia (50%), and South Africa (66%) having more than half of their population living in urban areas [29]. Most of this growth is due to rural–urban migration, the reclassification of rural areas, and the natural growth within cities. The movement of people, particularly from rural to urban areas, has led to an importation of poverty as the rural poor escape to the cities. With the majority of the world's population now living in urban areas, the urban share of poverty has also increased in a process that Ravallion et al. [30] have coined "the urbanization of poverty".

Using Namibia as a case study, we explored different scenarios of food insecurity under lockdown conditions. We postulated that lockdown COVID-19 conditions affect the urban food system, and, at a household level, this is manifested through limited food purchase frequency. For those households that depend on rural–urban food transfers, the restricted movement also does affect food access. Hence, it exacerbates the household food insecurity situation [31]. Studies conducted between 2008 and 2021 show a near constant food insecurity situation in Windhoek, one that became slightly worse during the pandemic restrictions. In 2008, food insecurity was 72%, $n = 1781$ [32], while in 2016, it was estimated at 74.6%, $n = 855$ [33], whereas in 2021, it was at 87.6%, $n = 3648$ [34]. Similarly, supermarkets and rural–urban food transfers have remained dominant sources of food [30,33]. The above endorses the assumption that the food insecurity landscape in Windhoek has not changed much since 2008. Thus, the same food supply chain and purchase behaviours have continued and remained similar during the time of the COVID-19 lockdown in 2020.

## 2. Methodology

### 2.1. Settings

The study was centered on three informal settlements in Katutura, Windhoek, Namibia. The informal settlements are Samora Machel Constituency, Tobias Hainyeko Constituency, and Moses Garoeb Constituency. The three informal settlements are located about 5 km from the city centre and are next to each other. These three informal settlements are densely populated (49.18, 45.91, and 41.99 inhabitants per square km, respectively) [35]. They are largely characterized by lack of basic infrastructure, high unemployment rates, poor water and environmental sanitation, poor housing, insecurity, violence, and poor health indicator.

Katutura has six constituencies [Tobias Hainyeko, Katutura central, Katutura East, Soweto (John Pandeni), Samora Machael, and Moses Garoeb], of which four (Tobias Hainyeko, Samora Machael, Moses Garoeb, and Khomasdal) have informal settlements. However, the Khomasdal constituency was not included in the survey. According to the 2011 Namibian Population and Housing Census, the total population in the settlements was 199,100 with 52,100 households and an average household size, per constituency, ranging between 3.3 and 4.9 persons. Therefore, over 60% of Windhoek's population lives in informal settlements. Figure 1 shows the map of Windhoek and the three informal settlements. The enumerated settlements are shown in Figure 2.

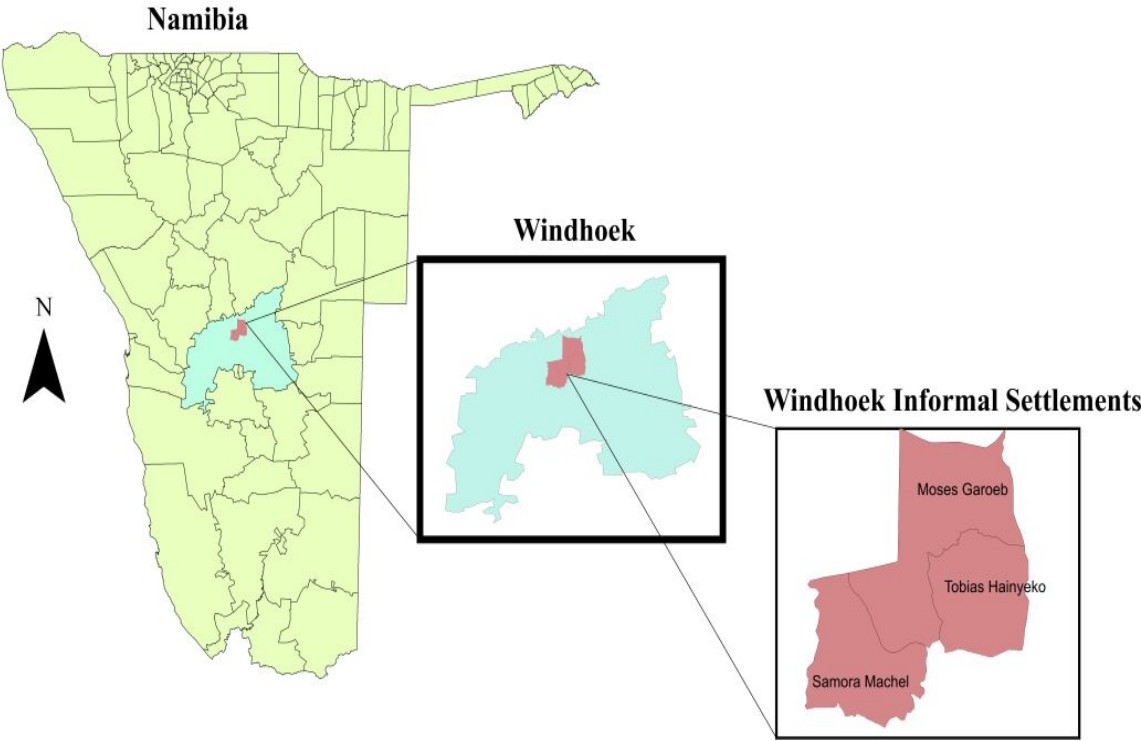

**Figure 1.** Map of Windhoek informal settlements. Source: [36].

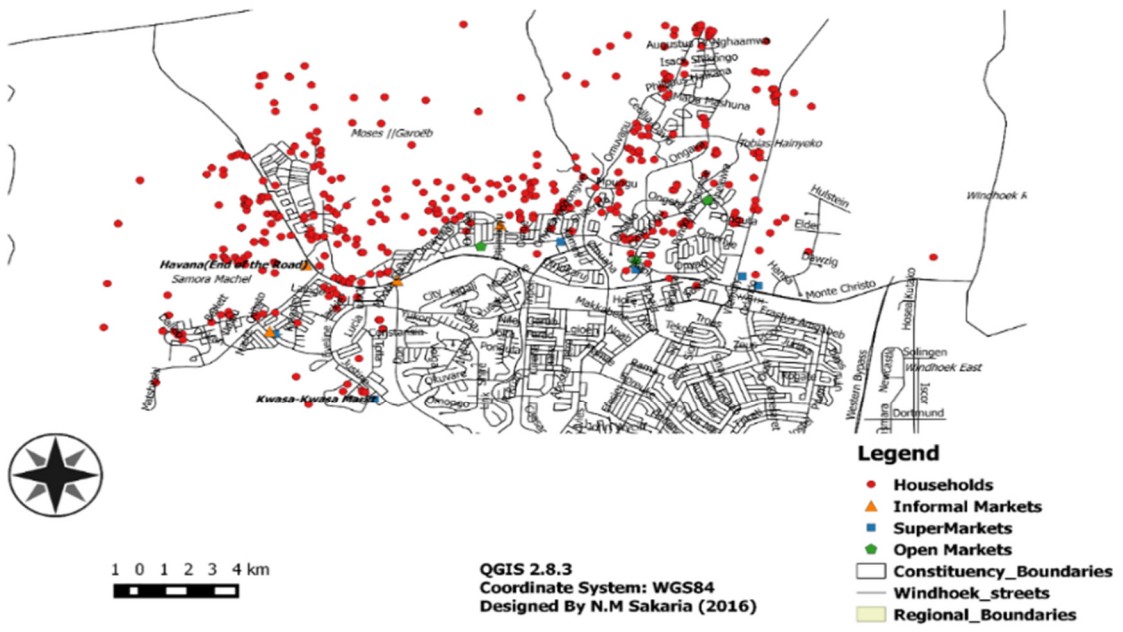

**Figure 2.** Enumerated informal settlements and households of Katutura, Windhoek. Source: [36].

*2.2. Data*

The data for this study was collected in 2016 by the Department of Statistics and Population Studies at the University of Namibia in partnership with the African Food Security Urban Network (AFSUN), the Hungry Cities Partnership (HCP), and the Balsillie School of International Affairs (BSIA). This survey used the AFSUN–HCP tool referred to as the Hungry Cities Food Purchase Matrix (HCFPM). The essence of this tool is that information

collected at household level (on the four dimensions of food poverty: availability, accessibility, utilization, and stability) can also be linked to macro-level livelihoods and urban food systems [37]. Overall, the survey collected a wide range of demographic, economic, and food consumption data. The food consumption data consisted of 32 commonly consumed food items, grouped into staples, fresh, packaged, frozen foods, and their sources (where household obtain their food and how often they patronize each source). Further, this included information of the households' experiential food insecurity and dietary consumptions. The survey used tablets to capture the data, and there was an in-built GPS that enabled the analysis to reflect where across the population poverty food insecurity existed.

A two-stage cluster sampling approach was used to pick households in the selected constituencies of Windhoek. Firstly, primary sampling units (PSUs) from a master frame developed and demarcated for the 2011 Population and Housing Census were randomly selected within the chosen constituencies of Windhoek. A total of 35 PSUs were selected, covering the whole of Katutura. Twenty-five households were systematically selected in each PSU, making a total sample size of 875 households. The sampled PSUs and households were located using maps, which were used to target households for interviews.

The sample size calculation for the cluster survey design was based on the following assumptions: (a) Target population size of 200,000 individuals or 300 clusters; (b) Estimated percentage in the target population with the event of interest (children malnutrition) of 25% based on the 2006/07 Namibian Demographic and Health Survey (NDHS); (c) Confidence interval width of 5% or Confidence coefficient of 95%; (d) Estimated Design effect (DEFF) of 2; (e) Percent Response of 60% (a combined effect of non-response and attrition); and (f) On average, there will be one eligible individual per household.

*2.3. Measures*

Most of the measures used in this study were constructed from the dataset. Detailed information on the derived variables is provided in Mbongo [36] and Nickanor et al. [38]. However, here we describe these variables in brief.

2.3.1. Food Insecurity Outcomes

The dependent variable, food insecurity prevalence (FIP), was a constructed outcome. FIP measures food inaccessibility and is constructed to take two levels (1 = insecure, 0 = secure), which is constructed as an index derived from a set of 10 questions on food accessibility [37]. Each of the 10 food access indicators has a four-level ordinal categorical response (0 = none, 1 = rarely, 2 = sometimes, 3 = often), with a high level indicating severe food inaccessibility. The 10 questions used are:

a. In the past four weeks, did you worry that your household would not have enough food?
b. In the past four weeks, were you or any household member not able to eat the kinds of foods you preferred because of a lack of resources (money)?
c. In the past four weeks, did you or any household member have to eat a limited variety of foods due to a lack of resources (money)?
d. In the past four weeks, did you or any household member have to eat some foods that you really did not want to eat because of a lack of resources (money) to obtain other types of food?
e. In the past four weeks, did you or any household member have to eat a smaller meal than you felt you needed because there was not enough food?
f. In the past four weeks, did you or any household member have to eat fewer meals in a day because there was not enough food?
g. In the past four weeks, was there ever no food to eat of any kind in your household because of lack of resources (money) to get food?
h. In the past four weeks, did you or any household member go to sleep at night hungry because there was not enough food?

i.　In the past four weeks, did you or any household member go a whole day and night without eating anything because there was not enough food?

j.　In the past week, did you or any household member eat a cooked meal less than once a day?

In this paper, the four HFIAP categories were binned into "food secure" and "food insecure," with 0 assigned to those in the "food secure" category, while 1 was assigned to those that fall in the three categories: "mildly food insecure," "moderately food insecure," and "severely food insecure".

### 2.3.2. Predictor Variables

The primary covariate is frequency of food purchase (FFP), which was reported as either daily, weekly, or monthly for all listed 32 food items. This considers both the source and type of food purchased. A second principal variable is the lived poverty index (LPI). The AFSUN–HCFPM used the LPI, a multidimensional indicator of experiential poverty. LPI is derived using sets of questions on how often a household has gone without certain basic goods and services, including food, medical care, fuel for cooking, and a cash income [39]. LPI scores ranges from 0.00 to 4.00, with a score of 4 or closer to 4 indicating more households "going without" those basic needs.

Other factors include socio-economic variables, such as income from waged work or business, the type of household (nuclear, extended, male-centered, or female centered), the type of housing structure (formal if permanent structure, or informal if temporary building materials were used), occupation, household size, informal work income (yes/no), and social grant support (yes/no). Table 1 provides the frequency distribution summaries of all variables used in the analysis.

**Table 1.** Frequency distribution of variables used in the analysis.

| Variable and Category | Number of Households | Percentage |
| --- | --- | --- |
| **Outcome variable** | | |
| Food insecurity: Yes | 719 | 83.6 |
| Food insecurity: No | 141 | 16.4 |
| **Main Predictor variable** | | |
| Frequency Purchase (Daily) | 8 | 0.9 |
| Frequency Purchase (Weekly) | 13 | 1.5 |
| Frequency Purchase (Twice Monthly) | 82 | 9.5 |
| Frequency Purchase (Monthly) | 757 | 88.0 |
| **Other Predictor Variables** | | |
| Housing type (Informal) | 476 | 55.9 |
| Housing type (Formal) | 375 | 44.1 |
| Household size: 1 member | 76 | 8.8 |
| Household size: 2–3 members | 256 | 29.8 |
| Household size: 4–5 members | 274 | 31.9 |
| Household size: 6 or more members | 254 | 29.5 |
| Household structure: Female-centered | 280 | 30.3 |
| Household structure: Male-centered | 163 | 19.3 |
| Household structure: Nuclear | 204 | 24.1 |
| Household structure: Extended | 201 | 23.7 |
| Household occupation: Formal | 401 | 50.7 |

**Table 1.** *Cont.*

| Variable and Category | Number of Households | Percentage |
| --- | --- | --- |
| Household occupation: Causal | 156 | 19.7 |
| Household occupation: Business | 89 | 11.3 |
| Household occupation: Others | 145 | 18.3 |
| Informal Work Income: No | 573 | 69.1 |
| Informal Work Income: Yes | 268 | 31.9 |
| Child/Pension/Disability Grant: No | 754 | 88.1 |
| Child/Pension/Disability Grant: Yes | 102 | 11.9 |
| Lived Poverty Index Score * | Mean = 1.31 | SD = 1.03 |
| Household Income: $\leq$NAD 700.00 | 139 | 21.7 |
| Household Income: 701.00–1500.00 | 142 | 22.1 |
| Household Income: 1501.00–2500.00 | 105 | 16.4 |
| Household Income: 2501.00–6300.00 | 128 | 19.9 |
| Household Income: 6301.00+ | 128 | 19.9 |

* Continuous variable summarized using mean and standard deviation (SD). The sample size varied across variables because of missingness.

### 2.3.3. Statistical Analysis

Our goal is to examine how food purchase frequency, when constrained as a result of COVID-19 containment through lockdowns, will impact food insecurity using household FIP. This analysis brings two statistical issues that must be dealt with at analysis level: sample selection and partial observability. In the sample selection, one variable will only follow when the other occurs or will be partially observed due to the occurrence of the other. For example, the construction of FFP permits to only observe households who selected daily, weekly, or monthly responses separately. Similarly, the experience of food insecurity will be observed depending on the purchase pattern, and this is exacerbated by the lockdown. Two approaches of quantifying this impact are presented through univariate and bivariate partial observability models.

### 2.4. Univariate Probit Model

We fitted predictive models by considering levels of food insecurity prevalence (food access) and frequency of food purchases the household faces. The univariate model is a simplification of the relation between FIP and FFP within the latent structure. We adopted a probit regression model with FIP as the response and FFP as the exposure variable, stratified by area of residence (informal or formal housing). We then adjusted for other socio-economic variables. In a probit model, the probability of being food insecure is given as [40,41]:

$$\Pr\left(y_{ji} = 1\right) = \Pr\left(y_{1i}^{*} > 0\right)$$
$$= \Phi\left(X_{ji}\beta_{j}; \rho\right)$$

where $\Phi(z)$ represents a standard normal cumulative distribution function (CDF) evaluated at $z$, with a correlation coefficient $\rho$. The expanded model incorporating predictor variables $(X_{ji}\beta_{j})$, is written as

$$\Phi(FIP[p_i]) = \beta_0 + \beta_1 FFP + \beta_2 Informal + \beta_3 Income + \beta_4 LPI + \beta_5 HHSize + \ldots \quad (1)$$

where $p_i$ is the probability of a household being food insecure, and $\beta_i$ are regression coefficients corresponding to the predictor variables. These will give a positive or negative estimate. The positive estimate suggests that the factor is more likely to be associated with

increased food insecurity, whereas a negative coefficient indicates the predictor is likely to reduce food insecurity in the household.

### 2.5. Bivariate Probit with Partial Observability

Partial observability in a bivariate probit model refers to the case where the response variable is the outcome of a paired decision or outcomes. In models as such, an outsider only observes the joint outcomes taken by the pair of actions without observing the individual actions. For example, a food insecurity situation will arise if various conditions of food access prevail, which may be a result of not being in a position to purchase food at an appropriate and usual time. If there is some movement restriction, such as disease containment through lockdown, then food purchase will not occur at the usual frequency and, hence, would lead to the reported food insecurity situation. Nonetheless, to an outsider, only the final outcome of food security or insecurity is observed. In this case, food security implies that food was both accessible and available, and that failure to access food could result in food not being available, ensuring food insecurity conditions and coping mechanisms.

Here, we use such a scenario to fit a joint model of food insecurity prevalence and frequency of food purchase through conditioning (if food purchase is restricted, then food insecurity will follow) to allow for partial observability.

The bivariate probit with partial observability model is defined as follows: Let $i$ denote the $i$th observation, which takes the values from 1 to $N$, $X_1$ be the covariate matrix of dimension $N \times k_1$, and $X_2$ be a covariate $N \times k_2$. Define the latent response for Stage 1 to be

$$y_{1i}^* = X_{1i}\beta_1 + \epsilon_{1i}$$

and Stage 2 to be

$$y_{2i}^* = X_{2i}\beta_2 + \epsilon_{2i},$$

such that $\beta_1$ and $\beta_1$ are the corresponding regression coefficients. Note that the stages do not need to occur sequentially. Define the outcome of the first stage to be $y_{1i} = 1$ if $y_{1i}^* > 0$ and $y_{1i} = 0$ if $y_{1i}^* < 0$. Similarly, define the outcome of the second stage to be $y_{2i} = 1$ if $y_{2i}^* > 0$ and $y_{2i} = 0$ if $y_{2i}^* < 0$. The error terms ($\epsilon_{1i}$, $\epsilon_{2i}$) are independently and identically distributed as multivariate normal with zero means $E(\epsilon_{1i}) = E(\epsilon_{2i}) = 0$, and unit variances $Var(\epsilon_{1i}) = Var(\epsilon_{2i}) = 1$, and correlation $\rho$.

The observed outcome is the product of the outcomes from the two stages

$$z_i = y_{1i}y_{2i}$$

Or, put differently

$$z_i = \begin{cases} 1 & if\ y_{1i}^* > 0,\ y_{2i}^* > 0 \\ 0 & otherwise \end{cases}$$

The joint probability of the various outcomes in a partial observability model is given by [40,41]

$$\Pr(Z_i = 1) = \Pr(y_{1i} = 1,\ y_{2i} = 1\ )$$
$$= \Phi_2(X_{1i}\beta_1,\ X_{2i}\beta_2; \rho)$$

$$\Pr(Z_i = 0) = \Pr(y_{1i} = 0\ \text{or}\ y_{2i} = 0\ )$$
$$= 1 - \Phi_2(X_{1i}\beta_1,\ X_{2i}\beta_2; \rho)$$

where $\Phi_2$ is a bivariate standard normal CDF.

The log–likelihood function is expressed as

$$L(\beta_1,\ \beta_2, \rho; Z) = \sum_{i=1}^{n} Z_i \ln[\Phi_2(X_{1i}\beta_1,\ X_{2i}\beta_2; \rho)] + (1 - Z_i) \ln[1 - \Phi_2(X_{1i}\beta_1,\ X_{2i}\beta_2; \rho)]$$

We now implement the bivariate probit with partial observability on the two joint outcomes, FIP and FFP. Unlike in the univariate model, where FFP was a predictor, here, it is jointly assumed as an outcome, which is assigned same or different set of predictors. The bivariate system in latent representation then can be expressed as

$$E(Y_{1i}|X_{1i}\beta_1) = \beta_{10} + \beta_{11} Income + \beta_{12} SexHH + \beta_{13} LPI + \beta_{14} HSize + \beta_{15} Occup + \beta_{16} Informal + \beta_{17} Htype \\ + \beta_{18} SGrant + \beta_{19} PoorHH + \epsilon_{1i}$$

$$E(Y_{2i}|X_{2i}\beta_2) = \beta_{20} + \beta_{21} Income + \beta_{22} SexHH + \beta_{23} LPI + \beta_{24} HSize + \beta_{25} Occup + \beta_{26} Informal + \beta_{27} Htype \\ + \beta_{28} SGrant + \beta_{29} PoorHH + \beta_{2,10} PlaceFP + \epsilon_{2i}$$

where $Y_{1i}$ and $Y_{2i}$ are FIP and FFP, respectively.

The models are fitted using *BiProbitPartial* package in R [41], using maximum likelihood estimation method. A more general representation can be found in [40].

## 3. Results

A total of 860 households are used in this study. Missingness amounted to less than 2% of the total sample, as a complete case analysis was used [42,43]. Food insecurity was widely prevalent in Katutura in 2016, with 84% of the interviewed households indicating a lack of adequate food in the last 30 days preceding the survey (Table 1). Most households were purchasing food once or twice a month (88% once a month and 9.5% twice a month, respectively). Very few reported weekly or daily purchases (Table 1). About 60% of the households were classified as informal, while most households had four or more members. A third of the households were female-headed. Two-thirds of the households were receiving an income of less than NAD 2500 per month, with 21% having a monthly income of NAD 700 or less. Considering the lived poverty index score, the surveyed households lacked, on average, two items, be it water, electricity, food, etc.

Figure 3 shows the relationship between food insecurity and food purchase frequency. Most food insecure households were likely to do monthly food purchases, while there was no difference between daily and weekly purchases for those who were food insecure. The pattern is similar for the food secure households, although the percentages were low.

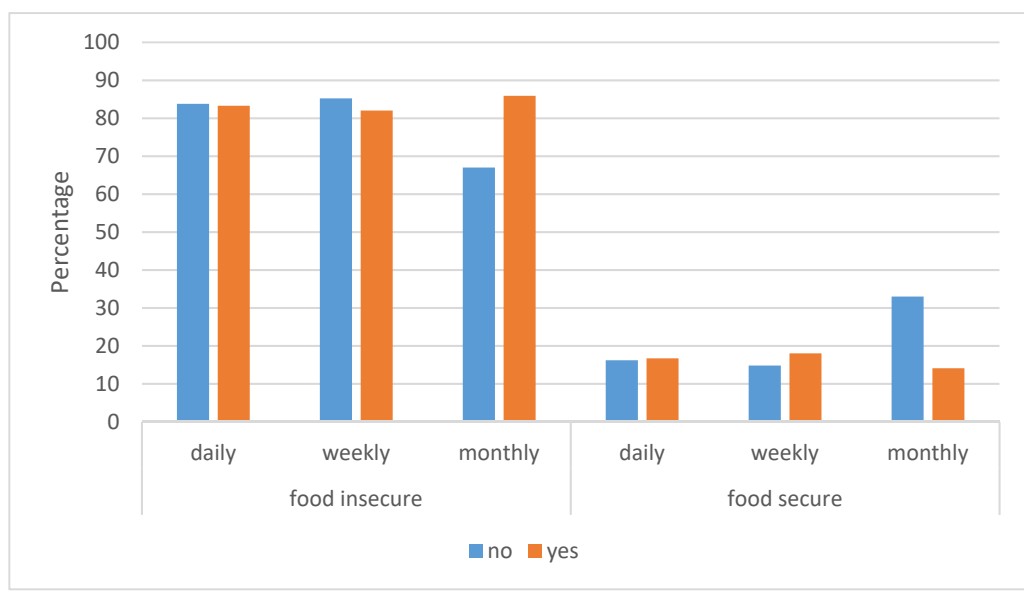

**Figure 3.** Clustered bar chart of food insecurity and food purchase frequency.

With regards to the association between FIP and FFP, Table 2 shows the results of their relation using a univariate probit model. Considering FFP alone we observe that daily food purchases were associated with an increased risk of food insecurity compared to those

who did so on a monthly basis (coeff = 0.076, $p$ = 0.05). Only those who were purchasing food twice a monthly were significantly associated with reduced food insecurity. The same pattern of association between twice monthly purchases is maintained in the full model. Other variables associated with food insecurity in the full model were housing informality, household size, household structure, LPI, and income.

**Table 2.** Results of predictors of food insecurity in Windhoek, Namibia using the univariate probit model.

| Variable | Basic Model | | | Full Model | | |
|:---:|:---:|:---:|:---:|:---:|:---:|:---:|
| | Coefficient | Std. Error | *p*-Value | Coefficient | Std. Error | *p*-Value |
| Intercept | 1.074 | 0.057 | 0.001 | 0.721 | 0.111 | 0.001 |
| Frequency Purchase (Daily) | 0.076 | 0.571 | 0.05 | −0.146 | 0.163 | 0.368 |
| Frequency Purchase (Weekly) | −0.572 | 0.368 | 0.61 | −0.039 | 0.198 | 0.844 |
| Frequency Purchase (Twice Monthly) | −0.699 | 0.153 | 0.001 | −0.201 | 0.045 | 0.001 |
| Frequency Purchase (Monthly) | 0 | . | . | 0 * | . | . |
| Housing type (Informal) | | | | 0.909 | 0.034 | 0.007 |
| Housing type (Formal) | | | | 0 * | . | . |
| Household size: 1 member | | | | −0.234 | 0.052 | 0.001 |
| Household size: 2–3 members | | | | −0.034 | 0.035 | 0.337 |
| Household size: 4–5 members | | | | 0.008 | 0.034 | 0.817 |
| Household size: 6 or more members | | | | 0 * | . | . |
| Household structure: Female-centered | | | | 0.016 | 0.039 | 0.682 |
| Household structure: Male-centered | | | | −0.095 | 0.044 | 0.034 |
| Household structure: Nuclear | | | | −0.094 | 0.041 | 0.021 |
| Household structure: Extended | | | | 0 * | . | . |
| Household occupation: Formal | | | | 0.003 | 0.044 | 0.953 |
| Household occupation: Causal | | | | 0.127 | 0.095 | 0.179 |
| Household occupation: Business | | | | 0.074 | 0.097 | 0.441 |
| Household occupation: Others | | | | 0 * | . | . |
| Informal Work Income: No | | | | 0.052 | 0.094 | 0.579 |
| Informal Work Income: Yes | | | | 0 * | . | . |
| Child/Pension/Disability Grant: No | | | | −0.007 | 0.048 | 0.878 |
| Child/Pension/Disability Grant: Yes | | | | 0 * | . | . |
| Lived Poverty Index score | | | | 0.084 | 0.014 | 0.001 |
| Household Income: ≤ NAD 700.00 | | | | 0.082 | 0.058 | 0.157 |
| Household Income: 701.00–1500.00 | | | | 0.121 | 0.049 | 0.013 |
| Household Income: 1501.00–2500.00 | | | | 0.116 | 0.054 | 0.022 |
| Household Income: 2501.00–6300.00 | | | | 0.121 | 0.047 | 0.010 |
| Household Income: NAD 6301.00+ | | | | 0 * | . | . |

* An entry with zero is the reference group.

For instance, those living in informal housing were more likely to be food insecure compared to those in formal housing (coeff = 0.909, *p*-value = 0.007), while small-sized households were less likely to be food insecure compared to large households (coeff = −0.234, *p* = 0.001). A similar pattern is obtained for male-centered and nuclear households, which show less food insecurity relative to those with extended types of households (coeff = −0.095, *p* = 0.034; coeff = −0.094, *p* = 0.021 for male-centered and nuclear households, respectively).

Food insecurity was also associated with the lived poverty index, with food insecurity increasing with LPI (*p* = 0.001). Table 2 also shows heightened food insecurity across different levels of income, particularly for those at income bands of NAD 701–NAD 1500 (*p*-value = 0.013), NAD 1501–NAD 2500 (*p*-value = 0.022) and NAD 2501–NAD 6300 (*p*-value = 0.01).

Table 3 presents a summary of a bivariate partial observability probit model. This result suggests that the unobservable variables influencing household food insecurity and frequency of purchase are correlated (r = 0.057, 95% CI: 0.0394, 0.085). This implies that household food insecurity increased with frequent food purchases, meaning daily or weekly purchases are associated with survivalist behaviour in the sampled households.

**Table 3.** Bivariate partial observability model on food insecurity and frequency of food purchase.

| Variable | Insec: Probability (Food Insecurity) | | | Purch: Probability (Food Insecurity \| Frequency of Food Purchase) | | |
|---|---|---|---|---|---|---|
| | Coefficient | Std. Error | *p*-Value | Coefficient | Std. Error | *p*-Value |
| Intercept | 1.114 | 0.7339 | 0.129 | 7.833 | 13.707 | 0.988 |
| Housing type (Formal) | 0 | . | . | 0 | . | . |
| Housing type (Informal) | 1.396 | 0.568 | 0.0142 | 1.173 | 0.392 | 0.069 |
| Household size: 1 member | 0 | . | . | 0 | . | . |
| Household size: 2–3 members | 2.517 | 0.652 | 0.00011 | −4.969 | 13.709 | 0.992 |
| Household size: 4–5 members | 4.245 | 0.841 | 0.00045 | −5.414 | 13.707 | 0.996 |
| Household size: 6 or more members | 4.082 | 0.902 | 0.00062 | −5.622 | 13.706 | 0.993 |
| Household structure: Female-centered | 0 | . | . | 0 | . | . |
| Household structure: Male-centered | −1.885 | 0.612 | 0.0021 | 0.524 | 0.486 | 0.281 |
| Household structure: Nuclear | −1.217 | 0.801 | 0.128 | −0.328 | 0.392 | 0.403 |
| Household structure: Extended | −1.905 | 0.744 | 0.011 | 5.192 | 6.468 | 0.927 |
| Household occupation: Formal | 2.693 | 1.524 | 0.132 | −5.327 | 7.380 | 0.992 |
| Household occupation: Causal | 4.368 | 5.732 | 0.446 | −5.944 | 7.386 | 0.991 |
| Household occupation: Business | 5.082 | 5.787 | 0.379 | −1.146 | 0.502 | 0.069 |
| Household occupation: Others | 0 | . | . | 0 | . | . |
| Informal Work Income: No | 0 | . | . | 0 | . | . |
| Informal Work Income: Yes | −4.371 | 5.734 | 0.445 | 0.699 | 0.757 | 0.355 |
| Child/Pension/Disability Grant: No | 0 | . | . | 0 | . | . |
| Child/Pension/Disability Grant: Yes | 10.111 | 61.327 | 0.978 | −1.125 | 0.515 | 0.028 |

The first part of the results shows the association of predictors with food insecurity, and the second part is demonstrated through the conditional equation. Here, we display the same type of predictors on the two equations since we cannot distinguish between variables that drive household food insecurity (**insec**) and variables that influence the frequency of household food purchases (**purch**). The general pattern that is emerging is that there are differences in coefficient signs between the **insec** and **purch** models, as well as the levels of significance. The **insec** model has many significant predictors compared to the **purch** equation. For instance, while informal housing is positive and significant for the **insec** model (coeff = 1.396, $p$-value = 0.0142), in the **purch** model, it is positive but marginally significant (coeff = 1.173, $p$-value = 0.069). This clearly suggest that the importance of the variable is somewhat the same when it comes to determining food insecurity and the frequency of food purchase.

Another thing that is emerging in the bivariate model is that there is a possibility of interaction between the **insec** and **purch** models. This can be seen in the association coefficients of household size and household structure, in that the predictors are highly significant in the **insec** model ($p < 0.001$ for all household size categories and $p < 0.001$ for male-centered household structures), which has improved compared to the univariate coefficients obtained earlier, as shown in Table 2. On the other hand, the same predictors are negative and not significant in the **purch** model (Table 3). With regards to occupation, we see the positive but not significant pattern of association in the **insec** model. However, in the **purch** model, the results are all negative and not statistically significant. This clearly underscores that the partial observability model allows for a more nuanced separation of alternative theoretical mechanisms influencing food insecurity. Households receiving a social grant were negatively associated with a frequency of purchase ($p$-value = 0.028), suggesting these are likely to have monthly purchases relative to nearly daily or weekly purchases.

We further considered if the metric variables, LPI, and income exhibit any nonlinear association with the probability of food insecurity and food purchase. Figure 4 presents the smooth functions for the **insec** model (top panel) and **purch** model (bottom panel). The effects of LPI and income in the **insec** and **purch** equations show different degrees of non-linearity. The probability of experiencing food insecurity is found to increase with LPI. The likelihood of food purchase also increases with LPI. Higher LPI is associated with an increased propensity of food insecurity and food purchase. Food access, as well as food purchases, appears to have a higher association with LPI than with income. Indeed, the point-wise confidence intervals of the smooth functions for income in the **insec** equation contain the zero line for the whole range of the covariate values. Similar conclusions can be drawn by looking at the **purch** model. The intervals of the smooth for income in the **purch** equation contain the zero line for most of the covariate value range. This suggests that income is a weak predictor of food insecurity and food purchase. However, the slight increasing pattern of income (middle line) suggests a probability of monthly purchases that is partially associated with income (bottom panel). It also reveals a diminishing likelihood of food insecurity with increasing income (top panel).

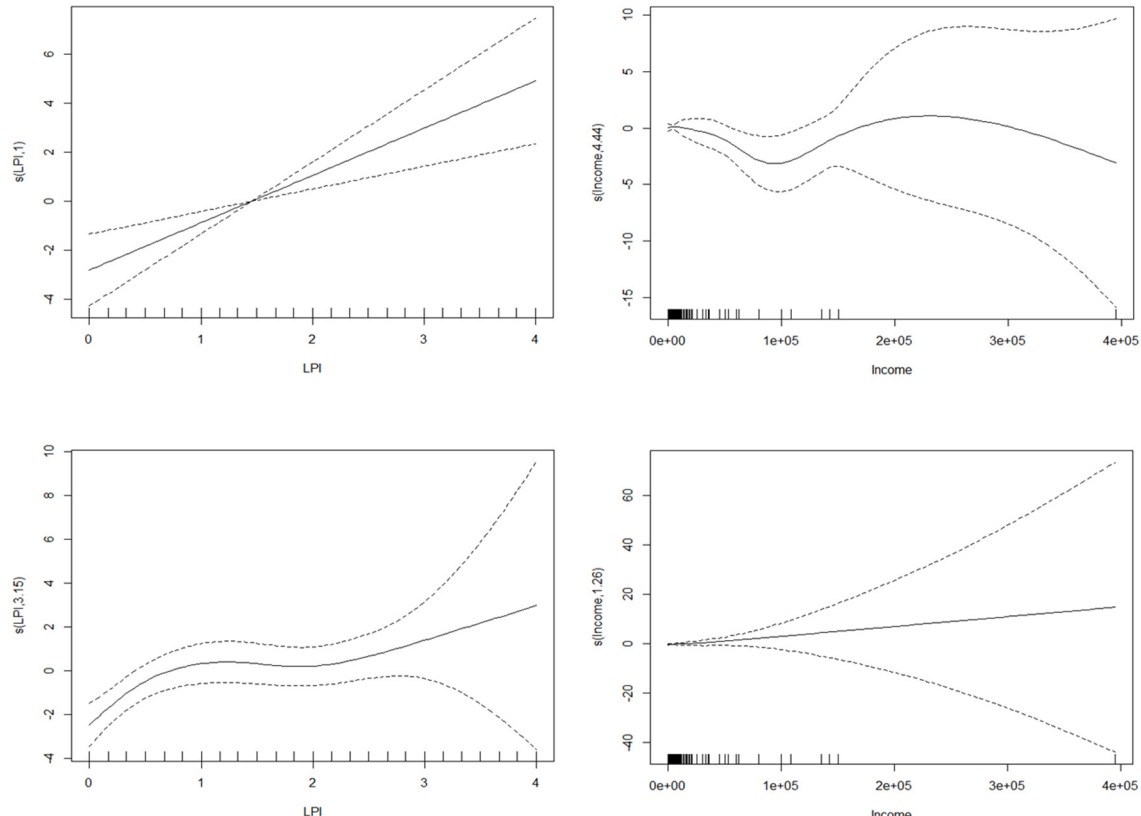

**Figure 4.** Nonlinear effects of LPI and income, with the estimated line provided as the solid central line accompanied by 95% confidence band, shown as dotted lines.

## 4. Discussion

The implementation of measures to reduce COVID-19 infection, such as physical distancing, working from home, travel bans, and other personal hygiene practices had affected, and to a large extent, disrupted the economy [15,27,44]. One of the impacts of the COVID-19 pandemic was the weakening of the community's economic condition. The COVID-19 pandemic had far-reaching effects, including food insecurity through disrupted food access [45] as a result of unavailability and a fluctuation in prices [46,47].

To assess these challenges and understand their impact on food security, this study considered the potential interaction between household food security and food supply systems in urban areas, using a case study of Windhoek, Namibia. The COVID-19 lockdown not only affected food availability and accessibility, but it also affected the stability of food supply and access. Similar studies have been conducted in many urban settings, for example, in Indonesia [47], Vermont, USA [45], and Jordan [48]. However, these only considered food insecurity and its associated socio-economic determinants. Here, we considered both the demand and supply sides of the urban food system, as both sides were severely impacted by the lockdown. We specifically considered the effect of frequency of food purchase (FFP) and subsequent food insecurity. Unlike earlier and similar studies [45,47,48], this study considered two-outcome joint models: one on food insecurity, and the other on the frequency of food purchase. On the probability of food insecurity, several factors were relevant, including FFP, informal housing, household size, household structure, LPI, and income. We observed that bi-monthly purchases were associated with lower food insecurity compared to monthly purchases. This indicates that the impacts of lockdowns are, first, likely to affect those who are food insecure (Figure 3) and may severely impact households if the lockdown is extended beyond two weeks (Tables 2 and 3). This approach clearly identified the behavioral and supply aspects of food security. It further

acknowledges the endogeneity associated with the purchase behaviour, which not only affects food security but is itself affected by other factors [49,50].

The probit and partial observability models yield identical results as far as the probability of food insecurity is concerned. However, the partial observability model reveals additional information about the effect of household size, household structure, and on the food insecurity–food purchase interaction by also allowing the estimation of the probabilities of food purchase influencing the state of food insecurity. We see, for example, that while the coefficients in the food insecurity model increased with most predictors, the food purchase coefficients decreased.

It is evident that each of these variables may affect the prevalence of food insecurity through three theoretical mechanisms, namely: an association with access, purchase frequency, and the place of purchase. The partial observability model permitted for evaluating each of the player-specific (FFP) and outcome-specific (FIP) latent components. This has important implications on the impact of the lockdown on food insecurity via the food purchase mechanism. This component is acting on both the food demand and supply mechanisms. A constraint on the food purchase will diminish the demand for food, which will in turn increase exposure to food insecurity. At the same time, the supply system is constrained, which may lead to food being less available and subsequently affect household food insecurity.

While this paper focused on the disruption with regards to purchase behaviour, the supply chain was affected by global supply. Unprecedentedly, disruptions in the global supply chain will affect vulnerable countries. Complete lockdowns, like those in India, South Africa, and Europe, have caused disruptions in the food supply chain, with the scarcity of labour making it even worse [51]. Moreover, most food supplying countries, had, from March 2020, imposed trade restrictions on exports. At least 17 countries sought to limit food exports to protect local supplies. On the other hand, other countries have accelerated the purchase of grains [52]. Such export-constraint actions have an influence on production, processing, transportation, and buying behaviour, which subsequently will have repercussions on volumes and prices of foods traded. Price fluctuations are likely, as prices may soar. Similarly, political factors, such as tensions, can undermine access to adequate food and the utilization of food. Implicit in this challenge is the link to dietary diversity. Consequently, it may affect nutrition security. Indeed, food security is a necessary condition for nutrition security but is not sufficient to equate it to nutrition security.

Heightened demand, disruption, and uncertainty threaten to produce a new global food crisis on the back of the outbreak, which could see further price hikes, food losses, and shortages, as well as rising malnutrition and global health issues in the months ahead [53]. Eventually, this could increase food insecurity and hunger, which will pose a challenge in meeting sustainable development goals. As food panic recedes, the IFPRI has moved to suggest that the global food supply system should be fixed, particularly in response to pandemic crises. Innovative measures can be deployed to minimize the impact of the virus on food security.

Strengthening the food supply system is essential to defending food and nutrition security and rural livelihoods in low-income households against the COVID-19 threat. Urban agriculture, in the case of Windhoek, is not an option. However, other avenues within the urban food systems need to be looked at. There are consequences of food access being limited to retail shops and shopping malls: the majority of Windhoek's households are poor. Those in the informal settlements, especially, rely heavily on the localized informal traders, namely open markets and street food vendors, for their food. Therefore, the interpretation of the COVID-19 regulations on lockdowns has resulted in the closure of local open markets and the banning of street food vendors, thereby depriving already food insecure households of food access.

Removing access to the informal trading options limits poor households shopping to centralized retail shops and shopping malls. This has implications for long-distance

traveling (transport) and queueing at shopping malls, which also increases exposure to COVID-19 and even greater security risks.

Furthermore, it is also unlikely that the informal traders will survive a long period of inactivity and will most likely close down. In the absence of localized markets and street food vending, the retail shops and shopping malls will dominate on food access, and these will have long-term ramifications for the economy of the informal settlements, as well as deepen the intensity of food-insecure households.

This study is not without limitations. First, the study used data from 2016 and assumed that the same conditions of food insecurity and purchase frequency will remain relevant at the time of the COVID-19 lockdown. While this might be a strong assumption, the socio-economic conditions have not changed much until today. A COVID-19 tracker study conducted by the Namibia Statistics Agency revealed that food insecurity was relatively high at a national average of 60.1% [34]. Future research is, therefore, warranted to capture the current food insecurity scenarios and purchase behaviour following the pandemic period. The second limitation is that the responses are self-reported. As such, there is a potential of biased reporting. However, since the selection of participants was random, we purport that any response bias was minimized.

## 5. Conclusions and Recommendations

The COVID-19 pandemic is undoubtedly one of the greatest challenges that the world has faced this century. With a very short incubation period, the disease is rapidly infecting many people within countries, forcing governments to adopt any measures that promise to give them respite and lessen the burden on health facilities. While the measures adopted are viewed as being largely necessary, not much focus is being directed towards understanding how the majority of the people are surviving under these health-driven measures. Considerations of food, a basic and necessary requirement for survival, are being relegated to the periphery of current policy debates and programming. Some of the health measures being put in place are impinging on the ability of households to access and procure food. We show in this paper that food insecurity manifestation is an interaction of several factors, some of which are embedded within the food supply system. Interfering with the functioning of the food system compromises food security through the frequency of food purchase. The analysis provided in this paper has presented just a partial aspect of this interaction and how food security is predicted to be in informal settlements. Thus, more nuanced types of models, for instance, through structural equation models will be worth it to explore the interactions from the food demand and supply while including all intermediary mediating factors.

**Author Contributions:** Conceptualization, L.N.K.; methodology, G.T., L.N.K. and N.M.N.; validation, L.N.K.; formal analysis, G.T. and L.N.K.; investigation, L.N.K. and N.M.N.; resources, G.T., L.N.K. and N.M.N.; data curation, L.N.K.; writing—original draft preparation, G.T.; writing—review and editing, L.N.K. and N.M.N.; visualization, L.N.K.; supervision, L.N.K. and N.M.N.; project administration, N.M.N.; funding acquisition, G.T. All authors have read and agreed to the published version of the manuscript.

**Funding:** This research was funded by African Food Security Urban Network (AFSUN).

**Informed Consent Statement:** Informed consent was obtained from all subjects involved in the study.

**Data Availability Statement:** Data is available from the authors on request.

**Conflicts of Interest:** The authors declare no conflict of interest.

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
