# Peer review of "The Threat of COVID-19 on Food Security: A Modelling Perspective of Scenarios in the Informal Settlements in Windhoek"

_land, doi:10.3390/land12030718_

Round 1

Reviewer 1 Report

Dear Authors, I have reviewed your manuscript entitled ' The threat of COVID-19 on food security: A modelling perspective of scenarios in the informal settlements in Windhoek '. The paper is well-written and makes a novel contribution to knowledge and practice. However, there are some issues that should be addressed to improve the quality of the paper. These issues are explained below.

1. Why is the interpretation variables different in the univariate probit model and bivariate partial observability probit model?

2. The variables that are not placed in the model in Table 2 can be displayed.

3. The specific meaning of the real line and curve in Figure 4 must be clearly identified in the figure.

4. The influence of probability and condition has been analyzed, so how big is the unconditional impact on the food insecurity?

5. The coefficients of probit model is not the direct effect. Generally, the impact of the explanatory variables on the explanation variable is generally displayed through marginal effects or elasticity.

6. Why not put income variable as a continuous variable in the model?

7. In order to avoid non-academic disputes such as the origin of the COVID-19 epidemic, it is recommended to delete the first sentence of the introduction 'A novel coronavirus responsible for COVID-19 respiratory disease was first reported 21 in Wuhan, China in December 2019  .'

Author Response

Response to Reviewers

Reviewer 1

Dear Authors, I have reviewed your manuscript entitled ' The threat of COVID-19 on food security: A modelling perspective of scenarios in the informal settlements in Windhoek '. The paper is well-written and makes a novel contribution to knowledge and practice. However, there are some issues that should be addressed to improve the quality of the paper. These issues are explained below.

1. Why is the interpretation variables different in the univariate probit model and bivariate partial observability probit model?

Response: Essentially the interpretation is similar, where if one gets a positive value this means an increased likelihood while a negative implies the opposite. Now to allows comparison between the two models, the results are presented together as to how they relate to the response variables. To permit similarity of interpretation, we have included in the text the coefficients, and corresponding p-values as presented in the univariate model.

2. The variables that are not placed in the model in Table 2 can be displayed.

Response: Thank you for this observation. We seem not to understand this comment. It should be noted all variables presented as summaries in Table 1 are also shown in Table 2, and further in Table 3.

3. The specific meaning of the real line and curve in Figure 4 must be clearly identified in the figure.

Response: We have expanded the caption of Figure 4 to allow clear identification of the estimated line and the corresponding 95% confidence band.

4. The influence of probability and condition has been analyzed, so how big is the unconditional impact on the food insecurity?

Response: This is noted. This is provided by the univariate model. The reults are given in Table 2. However, we have added in the bivariate model the level of correlation, using copula model, between food insecurity and food purchase frequency.

5. The coefficients of probit model is not the direct effect. Generally, the impact of the explanatory variables on the explanation variable is generally displayed through marginal effects or elasticity.

Response: Correct, this is particularly true in econometric models. Our aim is to capture association. However, the marginal effects can easily be obtained, but will go contrary to the objective of the analysis.

6. Why not put income variable as a continuous variable in the model?

Response: This has been done in the partial observability model. The results are provided in Figure 4.

7. In order to avoid non-academic disputes such as the origin of the COVID-19 epidemic, it is recommended to delete the first sentence of the introduction 'A novel coronavirus responsible for COVID-19 respiratory disease was first reported 21 in Wuhan, China in December 2019  .'

Response: This is observed. The sentence does not speak about origin, but rather where it was first reported, and two sources are provided. Besides these are known facts about where Covid-19 was first reported.

Reviewer 2 Report

This is a very interesting paper. The paper explored an approach to understand the potential impact of COVID-19 on household food security through a lens of purchase frequency by using data achieved before COVID-19. I have some suggestions for authors as follows.

As the HFIASS (Household Food Insecurity Access Scale) has a set of nine questions, I would suggest present the ten questions mentioned in line 182 and 183 or give a citation, or explain which question was added over the nine questions of HFIASS - “which is constructed by as an index derived from a set of ten question on food 182 accessibility”.

I think this sentence needs a citation- “LPI is derived using sets of questions 190 on how often a household has gone without certain basic goods and services including: 191 food, medical care, fuel for cooking and a cash income” (line 190-192).

Please explain how food insecurity prevalence (FIP) was generated/derived from the ten food access indicator has a four-level ordinal categorical response.

How housing type of formal and informal was identified?

As there could be different purchasing frequency for different type of food items, how frequency of food purchase (FFP) was determined for a household?

Author Response

Reviewer 2

Comments and Suggestions for Authors

This is a very interesting paper. The paper explored an approach to understand the potential impact of COVID-19 on household food security through a lens of purchase frequency by using data achieved before COVID-19. I have some suggestions for authors as follows.

As the HFIASS (Household Food Insecurity Access Scale) has a set of nine questions, I would suggest present the ten questions mentioned in line 182 and 183 or give a citation, or explain which question was added over the nine questions of HFIASS - “which is constructed by as an index derived from a set of ten question on food 182 accessibility”.

Response: A reference and the set of ten questions have been added.

I think this sentence needs a citation- “LPI is derived using sets of questions 190 on how often a household has gone without certain basic goods and services including: 191 food, medical care, fuel for cooking and a cash income” (line 190-192).

Response: A citation has been added. Mattes et al., 2016

Please explain how food insecurity prevalence (FIP) was generated/derived from the ten food access indicator has a four-level ordinal categorical response.

Response: Explanatory text added. lines 209-212

How housing type of formal and informal was identified?

Response: Explanatory text added. line 224-225.

As there could be different purchasing frequency for different type of food items, how frequency of food purchase (FFP) was determined for a household?

Response: Text added to explain.

Reviewer 3 Report

The manuscript has analyzed an interesting topic, but there are the following shortcomings.

1. In the abstract, the key findings should be briefly clarified.

2. The Keywords are missing in the manuscript, which must be added.

3. Many data used in the manuscript are relatively old, such as the confirmed Covid-19 cases, which sould be updated.

4. All the reference cited in the manuscript are before 2021, but actually, many ralated situations have changed. It is highly recommended to update the references, and present the latest situations, especially the recent research progress related to the subject of the manuscript.

5. On Page 2, it is mentioned that "There is limited literature on the socioeconomic determinants of the spread of infectious diseases in developed countries (Adda, 2016)". But actually, many literature had focused on the topic in 2021 and 2022, both in developed and developing countriesThe authors have significant lack of access to relevant recent literature. This flaw have reduced the possible originality and novelty of the manuscript. 

6. There is a lack of independent discussion about the originality and novelty of the research topic of the manuscript.

7. On page 3, it is mentioned that, "The study is centred on three informal settlements in Katutura, Windhoek, Namibia". The representativeness and research value of these three informal settlements have not been discussed.

8. On page 5, "food security" is mischaracterized as "food poverty".

9. On page 5, the description of the data only mentioned the data investigation and acquisition, but whether the data matches the research area of the manuscript and and how many samples have been actually used in the manuscript have not been discussed

10. The data for the manuscript was collected in 2016, but the COVID-19 has happened in the end of 2019. How the 2016 data was used to study the COVID-19 occurred in 2019?

11. According to Table 1, the number of households is 860, but on Page 5, it is mentioned that "making a total sample size of 875 households".

12. Due to doubts about the matching and description of the data used in the manuscript, it is difficult to judge the confidence of the empirical estimate results.

13. There is a lack of discussion about the limitations and possible future research directions of the manuscript. 

14. The specific content of the Supplementary Materials, Author Contributions, Funding, Data Availability Statement and Conflicts of Interest are missing.

Author Response

Reviewer 3

Comments and Suggestions for Authors

The manuscript has analyzed an interesting topic, but there are the following shortcomings.

1. In the abstract, the key findings should be briefly clarified.

Response: The abstract has been revised.

2. The Keywords are missing in the manuscript, which must be added.

Response: These have been added.

3. Many data used in the manuscript are relatively old, such as the confirmed Covid-19 cases, which should be updated.

Response: Revised accordingly

4. All the reference cited in the manuscript are before 2021, but actually, many ralated situations have changed. It is highly recommended to update the references, and present the latest situations, especially the recent research progress related to the subject of the manuscript.

Response: Current references added

5. On Page 2, it is mentioned that "There is limited literature on the socioeconomic determinants of the spread of infectious diseases in developed countries (Adda, 2016)". But actually, many literature had focused on the topic in 2021 and 2022, both in developed and developing countries. The authors have significant lack of access to relevant recent literature. This flaw have reduced the possible originality and novelty of the manuscript.

Response: Revised

6. There is a lack of independent discussion about the originality and novelty of the research topic of the manuscript.

Response: We appreciate the comment provided by the review, however, we feel the point raised is too broad. It could have helped if specific areas were mentioned. As it is the review seems to suggest there is no discussion at all. This does not assist in further improving the current discussion section.

7. On page 3, it is mentioned that, "The study is centred on three informal settlements in Katutura, Windhoek, Namibia". The representativeness and research value of these three informal settlements have not been discussed.

Response: lines 146-150 motivates the choice of these areas. The areas contain the majority (over 60%) of the population and thus provides a better picture of life in the informal settlements

8. On page 5, "food security" is mischaracterized as "food poverty".

Response: No food insecurity and lived poverty are different and this has been made clear in methods section on page 5. Moreover the text has been expanded and references given to provide more clarity to the readers.

9. On page 5, the description of the data only mentioned the data investigation and acquisition, but whether the data matches the research area of the manuscript and and how many samples have been actually used in the manuscript have not been discussed

Response: lines 173to 179 describe the data used in the study. Table 1 further provide a summary by key variables.

10. The data for the manuscript was collected in 2016, but the COVID-19 has happened in the end of 2019. How the 2016 data was used to study the COVID-19 occurred in 2019?

Response: The study present possible scenarios that would occur within the covid-19 restrictions, as the food insecurity landscape has not changed. A justification has been added on lines 131-135.

11. According to Table 1, the number of households is 860, but on Page 5, it is mentioned that "making a total sample size of 875 households".

Response: This is correct, as Table 1 is a cross-tabulation and so missing values will vary per set of cross-classification. In the end the analyis is based on complete case analysis.

The number of 875 was the calculated sample size. This is given on page 4. The analysis used data for 860 household. A sentence has been added in the Results section.

12. Due to doubts about the matching and description of the data used in the manuscript, it is difficult to judge the confidence of the empirical estimate results.

Response: Thanks to the reviewer for this comment. However, we are not sure what this matching means. Section 2.2 clearly describes the data sources and how these were obtained.

13. There is a lack of discussion about the limitations and possible future research directions of the manuscript.

Response: Added and Revised

14. The specific content of the Supplementary Materials, Author Contributions, Funding, Data Availability Statement and Conflicts of Interest are missing.

Response: Added

Round 2

Reviewer 1 Report

Accept in present form

Author Response

We thank the reviewer for their time and comments made towards improving the manuscript. The manuscript has been checked for to remove minor spelling issues as required.

Reviewer 3 Report

The author has made changes or responses to the comments in the revised version, some of which were reasonable, but the rest were basically with no persuasive changes or responses. The comments that are very important but still with no persuasive changes or responses are as follows.

1. Very few recent literatures have been added, and the review of the latest research still have significant omissions and flaws.

2. There is still no discussion about the originality and novelty of the research done in the manuscript.

3. On line 140-142, it's added that, "The assumptions is that the food insecurity landscape has not changed much since the 2008 and 2016 studies done by AFSUN (Nickanor et al., 2017), thus the same food supply chain continued during the time of Covid-19 lockdown in 2020". But actually, how could food insecurity remain unchanged for more than a decade in the same place? Is it possible? There is no discussion of the assumption which is clearly inconsistent with reality. The data for the manuscript was collected in 2016, but the COVID-19 has happened in the end of 2019. How the 2016 data was used to study the COVID-19 occurred in 2019? The assumption is the basis of the entire research of the manuscript, but the lack of reasonableness makes the credibility of all the conclusions of the manuscript questionable.

4. On line 250-251, it's mentioned that, "The sample size will vary across variables because of missingness". How do these missing data of variables affect the estimates? How are these missing data of variables handled during model estimation? No discussion was seen in the manuscript.

5. There are many errors in grammar, spelling, punctuation. The word order and choice of sentences are not authentic and do not conform to English writing habits. 

Author Response

The author has made changes or responses to the comments in the revised version, some of which were reasonable, but the rest were basically with no persuasive changes or responses. The comments that are very important but still with no persuasive changes or responses are as follows.

Response: We remain grateful to receiving further comments from the reviewer. This clearly shows colleguality required in publishing. This has assisted us in improving our manuscript further.  

1. Very few recent literatures have been added, and the review of the latest research still have significant omissions and flaws.

Response: Recent literature added, particularly in the discussion.

2. There is still no discussion about the originality and novelty of the research done in the manuscript.

Response: Revised. See lines 425-445.

3. On line 140-142, it's added that, "The assumptions is that the food insecurity landscape has not changed much since the 2008 and 2016 studies done by AFSUN (Nickanor et al., 2017), thus the same food supply chain continued during the time of Covid-19 lockdown in 2020". But actually, how could food insecurity remain unchanged for more than a decade in the same place? Is it possible? There is no discussion of the assumption which is clearly inconsistent with reality. The data for the manuscript was collected in 2016, but the COVID-19 has happened in the end of 2019. How the 2016 data was used to study the COVID-19 occurred in 2019? The assumption is the basis of the entire research of the manuscript, but the lack of reasonableness makes the credibility of all the conclusions of the manuscript questionable.

Response: We have revised to include the actual percentage of food insecurity in Windhoek, and added a recent result, obtained in 2021 from a national-wide survey that show little change in the food insecurity situation in the city, before and during the pandemic. Such figures should solidify the assumptions used in the study. See lines 130-139

4. On line 250-251, it's mentioned that, "The sample size will vary across variables because of missingness". How do these missing data of variables affect the estimates? How are these missing data of variables handled during model estimation? No discussion was seen in the manuscript.

Response: We have revised to include a sentence that missingness accounted for less than 2%, as such complete case analysis was used. Since missing data is not an issue, we did not include a discussion on it. However to guide reader, we have included two references to help how to handle missingness- in case if one would be tempted to force through imputation. See lines 333-335

5. There are many errors in grammar, spelling, punctuation. The word order and choice of sentences are not authentic and do not conform to English writing habits. 

Response: The errors have been corrected.